# Version 2 of the IASI NH$_3$ neural network retrieval algorithm; near-real time and reanalysed datasets

Martin Van Damme[1], Simon Whitburn[1], Lieven Clarisse[1], Cathy Clerbaux[1,2], Daniel Hurtmans[1], and Pierre-François Coheur[1]

[1]Université libre de Bruxelles (ULB), Atmospheric Spectroscopy, Service de Chimie Quantique et Photophysique, Brussels, Belgium
[2]LATMOS/IPSL, UPMC Univ. Paris 06 Sorbonne Universités, UVSQ, CNRS, Paris, France.

*Correspondence to:* Martin Van Damme (martin.van.damme@ulb.ac.be)

**Abstract.** Recently, Whitburn et al. (2016) presented a neural network-based algorithm for retrieving atmospheric ammonia (NH$_3$) columns from IASI satellite observations. In the past year, several improvements have been introduced and the resulting new baseline version, ANNI-NH$_3$-v2.1, is documented here. One of the main changes to the algorithm is that separate neural networks were trained for land and sea observations, resulting in a better training performance for both groups. By reducing and transforming the input parameter space, performance is now also better for observations associated with favourable sounding conditions (i.e. enhanced thermal contrasts). Other changes relate to the introduction of a bias correction over land and sea and the treatment of the satellite zenith angle. In addition to these algorithmic changes, new recommendations for post-filtering the data and for averaging data in time or space are formulated. We also introduce a second dataset (ANNI-NH$_3$-v2.1R-I) which relies on ERA-Interim ECMWF meteorological input data, along with surface temperature retrieved from a dedicated network, rather than the operationally provided Eumetsat IASI L2 data used for the standard near-real time version. The need for such a dataset emerged after a series of sharp discontinuities were identified in the NH$_3$ timeseries, which could be traced back to incremental changes in the IASI L2 algorithms for temperature and clouds. The reanalysed dataset is coherent in time and can therefore be used to study trends. Furthermore, both datasets agree reasonably well in the mean on recent data, after the date when the IASI meteorological L2 version 6 became operational (30 September 2014).

## 1  Introduction

Ammonia measurements from space have come a long way since the first observations were reported (Beer et al., 2008; Coheur et al., 2009). It is now globally, and routinely measured with the main hyperspectral infrared sounders in orbit: TES, IASI, AIRS and CrIS (Shephard et al., 2011; Whitburn et al., 2016; Warner et al., 2016; Shephard and Cady-Pereira, 2015). For the retrieval of column abundances, two main approaches are followed. Iterative retrievals are based on fitting a calculated spectrum onto the observed spectrum. These can include the use of a priori information and typically provide a comprehensive uncertainty budget characterisation. They have the disadvantage of being computationally demanding as for a single retrieval a forward model has to be run several times.

The atmospheric spectroscopy group at ULB has developed several retrieval approaches based on the conversion of spectral $NH_3$ indices. These are indices that quantify the magnitude of the $NH_3$ absorption/emission lines in the spectrum. First Brightness Temperature Differences (BTDs) were used (Clarisse et al., 2009), later Hyperspectral Range Indices (HRIs) (Van Damme et al., 2014a). These type of methods rely on the fact that the indices can be converted to a column by taking into account the spectral sensitivity to the $NH_3$ abundance in the observed scene. For low to moderately high columns, both BTDs and HRIs are correlated linearly to column abundances, with the conversion factor depending on the thermal contrast (Van Damme et al., 2014a) and a host of other parameters. HRIs are derived from linear retrievals using a constant gain matrix which includes a generalised error covariance matrix. Background and full discussion of this index can be found in the following papers: Walker et al. (2011, 2012); Clarisse et al. (2013); Van Damme et al. (2014a); Whitburn et al. (2016). Van Damme et al. (2014a) used 2D look-up tables to convert HRIs to columns, while Whitburn et al. (2016) used a neural network (NN) to perform the conversion. The advantage of the latter is that it allows a much larger number of input parameters to be taken into account. It has a number of significant other advantages, which are outlined in the original paper.

In the first part of the present paper we report and detail several improvements that have been introduced to the original neural network based retrieval, here referred to as 'Artificial Neural Network for IASI'-$NH_3$-v1 (ANNI-$NH_3$-v1). In addition, we formulate a new set of recommendations on how to post-process, treat and interpret the data. In the final part we introduce a new dataset ANNI-$NH_3$-v2.1R-I, which differs from the baseline dataset ANNI-$NH_3$-v2.1 in that it uses different input data. While our baseline version uses operationally provided meteorological level 2 (L2) data, this reanalysed dataset relies on input data from the ERA-Interim ECMWF reanalysis (Dee et al., 2011) and a secondary neural network for surface temperature retrieval. The need for such a dataset arose after discontinuities were found in the analysis of timeseries which could be traced back to version changes in the IASI L2 processing chain for temperature and clouds. This new self-consistent dataset is introduced and detailed in Section 3.

## 2  The baseline version (ANNI-$NH_3$-v2.1)

### 2.1  Neural network setup and training

Thermal Contrast (TC) is a key quantity for infrared remote sounding of the lower troposphere (Clarisse et al., 2010; Bauduin et al., 2017). It is defined as the temperature difference between the surface and the air at a given altitude. Here we calculate the TC with respect to the 500 m air temperature (note that in Whitburn et al. (2016) it was defined with respect to the 1.5 km air temperature). The ANNI-$NH_3$-v1 has a rather poor performance for observations with a thermal contrast larger than 10 K, both above land and sea (see Figure 1). This is somewhat surprising as exactly the opposite would be expected. A first reason for this behaviour is that such high TC are under-represented in the v1 training dataset. To remedy this, a large amount of simulations were added to the training set, in such a way that a uniform distribution in terms of TC was achieved. Because TC varies much more over land than over sea, it was decided to make a separate training set (and neural network) for land and sea scenes. Additionally, it was observed that lower latitudes were under-represented in the sea training set due to the way the atmospheres were selected (from random IASI observations, for which the higher latitudes are overrepresented due to the

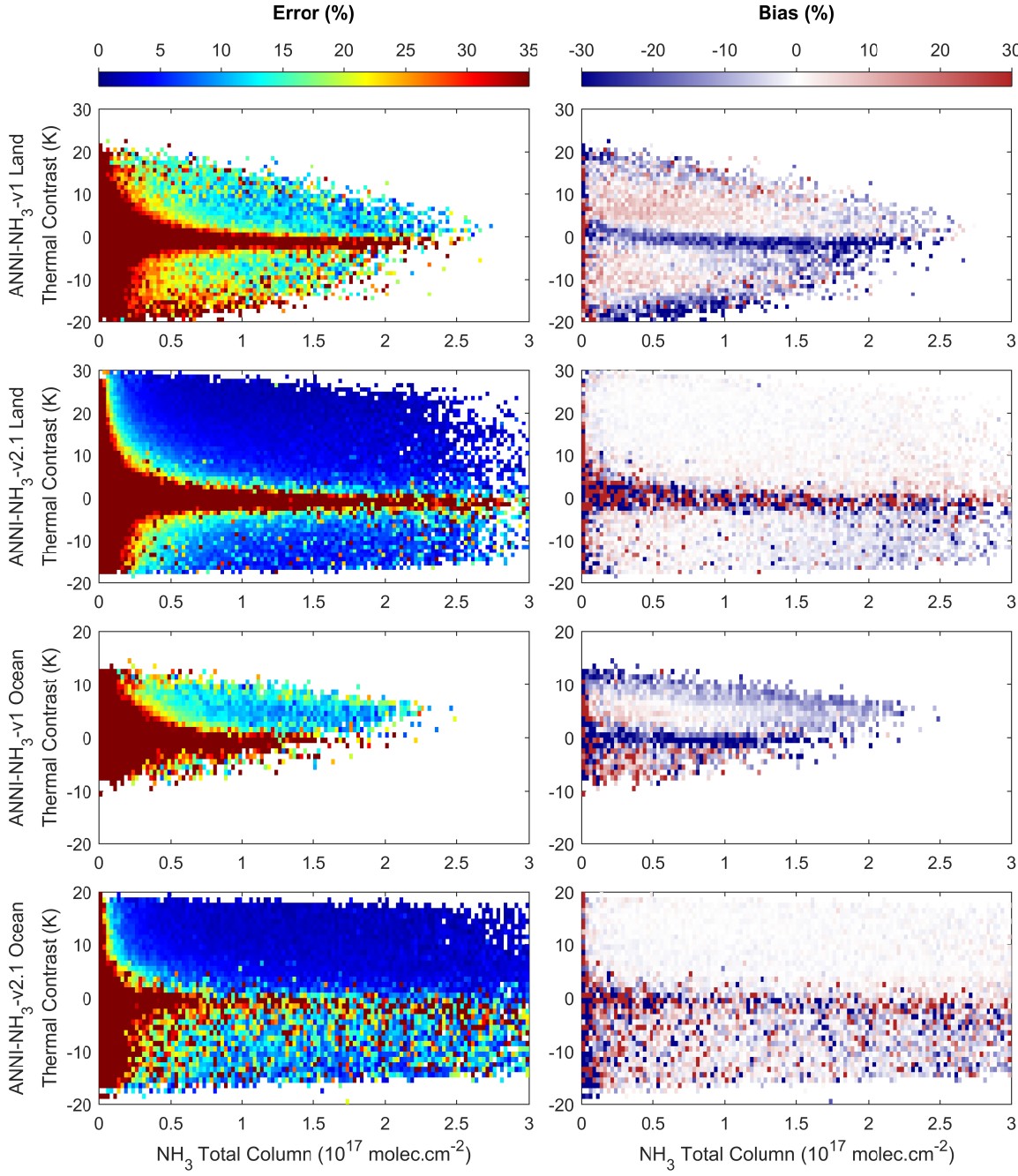

**Figure 1.** Neural network training performance in terms of mean error (left, %) and bias (right, %), for land (first two rows) and sea (bottom two rows) and for ANNI-NH$_3$-v1 (first and third row) vs ANNI-NH$_3$-v2.1 (second and fourth row).

Metop polar orbit). And so, over 36000 simulations were added for the lower latitudes. All in all, the complete training dataset is now almost double compared to the previous set, with around 450000 simulations (273000 over land and 172500 over sea).

Another reason for the relatively poor performance of the higher TC observations in ANNI-$NH_3$-v1 is related to the way the neural network was setup. As explained in Whitburn et al. (2016) the output variable of the neural network is not the total column of $NH_3$, but rather the ratio of the column, $[NH_3]_{v1}$ to the HRI. The retrieved column is thus the (observation dependent) ratio $f$ multiplied by the HRI. Thus we have

$$\text{output}_{v1} = f_{v1}(\text{input}_{v1}) = \frac{[NH_3]_{v1}}{\text{HRI}_{v1}} \rightarrow [NH_3]_{v1} = \text{HRI}_{v1} \times f_{v1} \tag{1}$$

The main rationale of using a ratio is that the neural network can be trained on noise free data and that the instrumental noise is propagated to the column in a transparent (linear) way. There is one catch though: for scenes with almost no sensitivity to $NH_3$ (therefore HRI $\approx 0$ ), the ratio can assume very large values, which can be problematic for training the network properly. To see this, first note that as the ratio is large and the sensitivity poor, the absolute error of the output will tend to infinity. The total cost function, defined as the mean squared error of the training dataset will therefore be dominated by these. The end result is that the part of the training set corresponding to HRIs very close to zero leads to non-convergent training or a badly performing network. In Whitburn et al. (2016), this situation was remedied by excluding those observations of the training set with a ratio larger than $7 \cdot 10^{16}$ molec.cm$^{-2}$. However, the fact that observations with a poor sensitivity (lower HRI for a constant $NH_3$ column) have a higher weight in the training cost function than those with a high sensitivity still makes the training focus on that part of the training set with the lowest sensitivity. This would not be a problem if one can train the neural network perfectly for the complete input space, but this is clearly not the case. In the new version we have reversed the ratio on which the training is performed:

$$\text{output}_{v2} = f_{v2}(\text{input}_{v2}) = \frac{\text{HRI}_{v2}}{[NH_3]_{v2}} \rightarrow [NH_3]_{v2} = \frac{\text{HRI}_{v2}}{f_{v2}} \tag{2}$$

Therefore, with this change, observations with an associated good sensitivity to $NH_3$ should be trained much better than before. The performance might get worse for the ones associated with a poor sensitivity, but these observations already have large uncertainties. Figure 1 shows the actual neural network training performance of both ANNI-$NH_3$-v1 and ANNI-$NH_3$-v2.1. In this figure, note that land and sea cases have been separated for v1, even though a single neural network was used to retrieve $NH_3$. As in Whitburn et al. (2016), these plots are representative for real observations since they include the most important observational error (the uncertainty on the HRI). The theoretically expected performance improvements outside the blind spot region of TC$\approx 0$ are evident. In the v1, the average training error started from 15 % ; while in the v2, this drops to around 5 %. Also, the biases mostly disappear: v1 was biased both inside and outside the blind spot. Note finally that these plots also demonstrate the much larger range of TCs covered by the v2.

The overall training of the v2.1 network is largely improved thanks to this simple change of output variable outlined above; however other changes also played a role in improving the performance on the training dataset. The main one is the addition of the HRI as an input parameter. While the ratio $f$ is independent of the column in the linear regime, linearity is only valid for low to medium columns. The departure from linearity can actually be observed as gradients in the v1 bias plots of Figure 1.

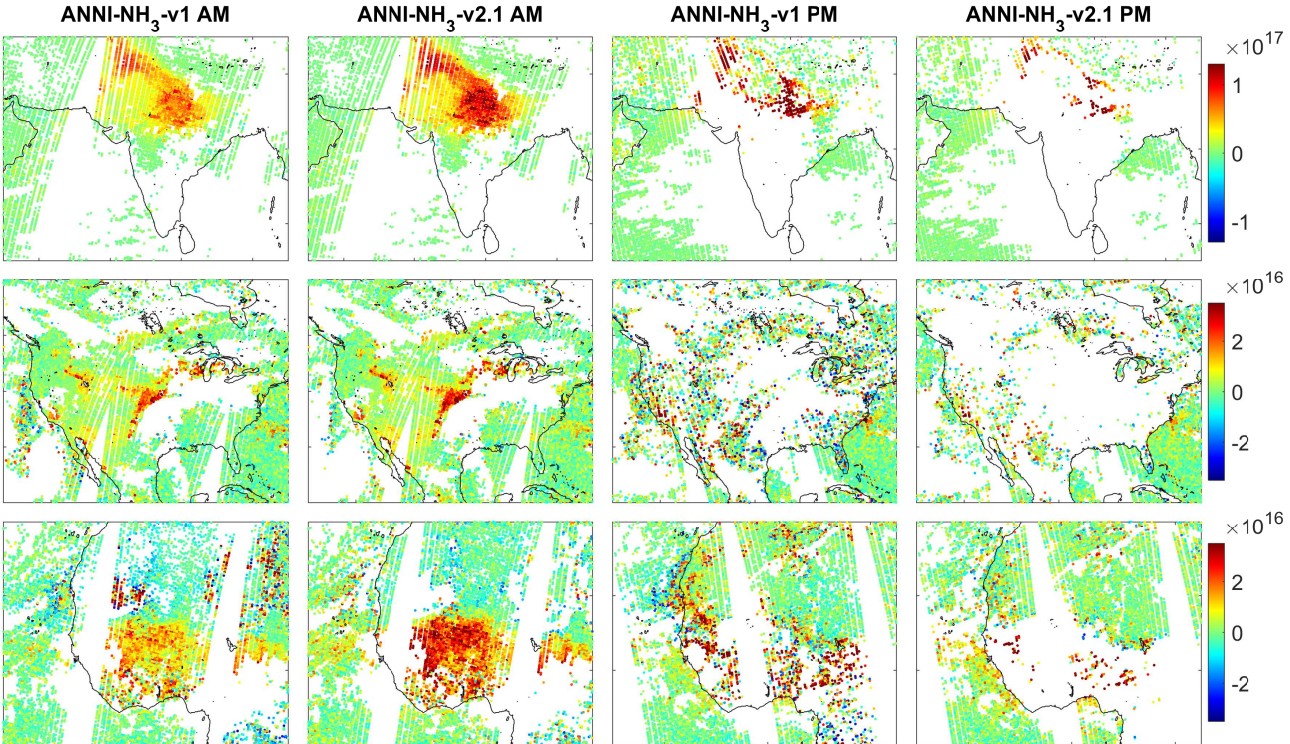

**Figure 2.** Example retrievals of the NH$_3$ column (molec.cm$^{-2}$) on 17 June 2015 for ANNI-NH$_3$-v1 (first and third column) and ANNI-NH$_3$-v2.1 (second and fourth column), morning (AM, left two columns) and evening overpass (PM, right two columns), over South Asia (top row), North America (middle row) and the western part of North and Central Africa (bottom row).

Adding the HRI as an input parameter allows the NN to correct for this. A number of input parameters have also been removed in v2, to keep the network as simple as possible and to avoid over-fitting. In particular, the input vertical profiles of H$_2$O and the pressure have been replaced respectively by a single H$_2$O total column and the surface pressure. With these changes the total number of input parameters is now reduced from 35 in v1 to 20 in v2. Table 1 lists these parameters and summarizes the
5  changes between NN-v1 and NN-v2.1.

Another change introduced in v2, which also contributes to making the NN simpler, is the way the viewing angle of the satellite is taken into account. In v1, angle dependent error covariance matrices (and therefore HRIs) were used following Bauduin et al. (2016). The reason for this was that earlier experience had shown that a straightforward angle correction on the airmass can result in biases on the final column (Van Damme et al., 2014a). In v2.1 the angle problem has been re-evaluated
10  after it emerged that the columns in v1 still showed an angle dependence, especially noticeable for the larger angles. This could for instance be seen by looking at the correlation between angle and total column over a small area over one season. The reason for this is unclear, and having effectively many different HRIs to train makes it difficult to trace the problem.

**Table 1.** List of changes from ANNI-NH$_3$-v1 to ANNI-NH$_3$-v2.1.

| | NN-v1 | NN-v2.1 |
|---|---|---|
| Output parameter | $\dfrac{[NH_3]}{HRI}$ | $\dfrac{HRI}{[NH_3]}$ |
| Input parameters* | 31: T (12 levels), $T_{surf}$, P (11 levels), $H_2O$ (7 levels), $\sigma$, $z_0$, $\epsilon$, angle | 20: T (12 levels), $T_{surf}$, $P_{surf}$, $H_2O$ total column, $\sigma$, $z_0$, $\epsilon$, angle, HRI |
| Training set | 250000 simulations | 450000 simulations |
| Land/sea treatment | One network | Separate networks |
| Angle treatment | Angle dependent HRIs | $1^{st}$ order correction of the HRIs by the cosine of the zenith angle |
| | | Angle as input parameter for $2^{nd}$ order corrections |
| Bias correction | No | Over sea (v2 dataset) / Over land and sea (v2.1 dataset) |
| Pre-filtering** | Cloud cover $> 25\,\%$ | Cloud cover $> 25\,\%$ |
| Post-filtering** | $[NH_3] < 0$ and HRI $> 1.5$ in absolute value | $[NH_3] < 0$ and HRI $> 1.5$ in absolute value |
| | $\dfrac{[NH_3]}{HRI} > 3 \cdot 10^{16}$ molec.cm$^{-2}$ in absolute value | $\dfrac{[NH_3]}{HRI} > 1.75 \cdot 10^{16}$ molec.cm$^{-2}$ in absolute value |

*$\sigma$ and $z_0$ are parameters characterizing the shape of the NH$_3$ vertical profile; $\epsilon$ represents the emissivity.

**An observation is removed as soon as one of the criteria is met.

For ANNI-NH$_3$-v2.1 we therefore decided to adopt again the approach introduced in Van Damme et al. (2014a) of simply correcting the HRI by the cosine of the zenith angle. An HRI consists of two components, the NH$_3$ signal component and a noise component. Clearly, the cosine correction only makes sense on the signal component, but applying the correction on the entire HRI value should in principle not cause any biases, apart from a compression of the instrumental noise for large

angle observations. The reason why this approach caused the introduction of biases in Van Damme et al. (2014a) lies in the fact that in that retrieval only positive values were retrieved. Especially for retrievals dominated by noise, applying a cosine factor will lead on average to lower values for larger angles. The neural network retrieval scheme maps the instrumental noise, which is on average one HRI unit, symmetrically around 0 to the NH$_3$ column space; so that no such bias effect is expected. We now therefore decided to correct the HRI value with a cosine factor, prior to feeding it to the network. This HRI

is the same as introduced in Van Damme et al. (2014a), for which the error covariance matrix was built using observations with all possible viewing angles. Note that the zenith angle is still kept as a parameter in the neural network, to allow the neural network to perform second order corrections to address any remaining angle dependency. The v2.1 angle correction was deemed satisfactory after analysis over different land and sea regions, at different times of the year.

     A final change introduced in v2 is a HRI bias correction over the seas, where the HRI was found to be slightly negative

overall, and decreasing with increasing H$_2$O total column amount. As this was identified to be also the case over land, the same correction was applied over land in v2.1. A H$_2$O dependent bias was determined from a region assumed NH$_3$-free by

calculating the median over sea for 30 days in 2015 over bins of $0.1 \cdot 10^{23}$ molec.cm$^{-2}$ of H$_2$O total column. These median values are then used to correct the HRIs before using them as an input in the neural network.

## 2.2 Performance on real data and recommendation for use

In Whitburn et al. (2016), a post-filter was applied to the retrieved data to remove the unphysical measurements (e.g. large negative columns associated with a large positive HRI). For ANNI-NH$_3$-v2.1, we have extended this post-filtering process to remove more of the blatantly erroneous retrievals. The current filtering procedure removes the observations for which:

1. The cloud coverage exceeds 25 %

2. The column is negative and HRI is larger than 1.5 in absolute value

3. $\dfrac{[\text{NH}_3]}{\text{HRI}}$ is larger than $1.75 \cdot 10^{16}$ molec.cm$^{-2}$ in absolute value

The first two criteria were already present in this form in v1, but the third criterion was weaker.

Example NH$_3$ total column retrievals are shown in Figure 2 for IASI morning and evening overpasses on 17 June 2015 over South Asia, North America and the western part of North and Central Africa for both v1 and v2.1 datasets. Overall it can be observed that both retrievals highly correlate, in particular, that the elevated and background columns occur in the same places. One noticeable difference is the effect of the extended post-filter in v2, which removes more of the larger negative columns over sea and over land on the evening overpass. Looking at the evening overpass over India it could be argued that the filtering is too aggressive. However, the observations that were removed there, are associated with extremely large uncertainties due to an almost zero sensitivity (TC very close to zero). For these observations that were removed, all that one can realistically say is that the NH$_3$ columns are enhanced. The observations over Africa on the other hand suggest that the post filtering procedure is still not strict enough. The current post-filtering flags were set by looking at a lot of different scenes from different parts of the year, and we consider them to be reasonable. The second important difference is that v2.1 columns are larger than v1 over the source regions (on average about 20 % for TCs above 10 K and columns higher than $1 \cdot 10^{16}$ molec.cm$^{-2}$). It is worth noting that the uncertainty associated with v1 and with v2.1 does not vary substantially for source regions, as it is mainly driven by the HRI and the temperature profile.

Measurements of NH$_3$ from space have a very large variability in their associated uncertainty, due to the variable sensitivity of the infrared outgoing radiation to the lower troposphere, as determined primarily by the TC (Clarisse et al., 2010; Bauduin et al., 2017). Summer-daytime is typically the best time to measure ammonia, while nighttime and/or winter are the worst, but the sensitivity can vary greatly even from one day to the next. Uncertainty estimates on the column range from 5 % to over 1000 %. Averaging such heterogeneous data is in general problematic, as the average can very easily be dominated by the data with the largest uncertainty, rendering the 'average measurement' meaningless. Van Damme et al. (2014b, 2015); Whitburn et al. (2015) therefore employed weighted averages, where the measurements with lowest uncertainty have the most weight in the average. Still, there are many approaches to weighting, and there is no uniquely best way of doing it. For a given TC, a larger column will always have a smaller relative error than a smaller column, so that weighting measurements with the

**Table 2.** Updates to Eumetsat's operationally distributed IASI L2 temperature and cloud products.

| Release date | Version | Comment |
|---|---|---|
| 27 Nov 2007 | 4.0 | Initial release of IASI/Metop-A L2, provided for even pixels only. |
| 29 Apr 2008 | 4.2 | Major changes in cloud coverage, surface temperature and temperature profiles. |
| 12 Aug 2008 | 4.3 | |
| 21 Jan 2009 | 4.3.2 | Surface temperature only provided for the cloud free observations. |
| 3 Mar 2010 | | L2 provided for both even and odd IASI pixels. |
| 29 Mar 2010 | 4.3.3 | |
| 14 Sep 2010 | 5.0.6 | Improved T profiles, but available for fewer observations. From this version onwards, temperature profiles and surface temperatures are provided for the same observations. Increased number of cloud free observations. |
| 2 Dec 2010 | 5.1 | Temperature information is now also provided for cloudy pixels (more than half of the IASI observations now have this info). |
| 20 Oct 2011 | 5.2.1 | Improved cloud screening for T retrievals. |
| 28 Feb 2012 | 5.3 | Major change in the cloud detection algorithm resulting in a decrease of the number of cloud free observations. Temperature information is now provided for observations with a cloud coverage below 25 %. |
| 16 July 2012 | 5.3.1 | |
| 8 Mar 2013 | | Initial release of IASI/Metop-B L2. |
| 30 Sep 2014 | 6.0.5 | Major update in the processing algorithm with the arrival of a new IASI L2 processor. IASI meteorological L2 data are now provided for nearly all IASI observations. |
| 24 Sep 2015 | 6.1 | Updates to the surface temperature algorithm. |
| 4 May 2016 | 6.2 | Important improvements to the temperature retrieval algorithms. |

relative error will always bias the result high (similarly, weighting with the absolute error tends to bias the result low). We refer to Whitburn et al. (2016) for examples and a discussion on the pros and cons of weighting measurements. With the extended post-filtering introduced in v2, we no longer recommend using weighted averages. Instead, if averages have to be performed we would now suggest to use unweighted averages. Using median values is also an option allowing to decrease the importance of outliers. However, it is always better wherever possible to use the individual measurements with their associated uncertainty, and avoid averaging.

## 3 An ERA-Interim reanalysis (ANNI-NH$_3$-v2.1R-I)

Up to now, for all our retrieval algorithms, we have only relied on IASI L2 meteorological data (August et al., 2012) to be used as input data (see e.g. Clarisse et al. (2012); Hurtmans et al. (2012); Van Damme et al. (2014a)). The relevant L2 data consist of the surface temperature, surface pressure, temperature and water vapour vertical profiles and cloud coverage

fractions. Since the first L2 meteorological data were operationally disseminated in 2007, a series of updates have been released, of which the most relevant are summarised in Table 2. Some of the updates led to changes in L2 data availability. This is illustrated for land observations over the Northern Hemisphere in the bottom panel of Figure 3. For instance, before 3 March 2010, L2 data were only provided for one in two pixels. Also, between January 2009 and up until October 2010, surface

temperature was only provided for observations with a 0 % cloud coverage. Other changes directly improved the quality of the L2. The NOAA PROducts Validation System (NPROVS, www.star.nesdis.noaa.gov/smcd/opdb/nprovs/NPROVS_trends.php) demonstrates convincingly that the agreement of L2 temperature profiles with radiosonde data improves markedly over time, both in the standard deviation and in the bias. The largest jump in improvement came after the introduction of IASI L2 v6 on 30 September 2014, but also with v6.2 on 4 May 2016.

The analysis of ANNI-NH$_3$-v2.1 timeseries revealed several rather sharp discontinuities which seemed to coincide with IASI L2 version changes (see top panel of Figure 3). In particular, a noticeable overall increase in the NH$_3$ columns was found to correspond with the v5 to v6 change, and a smaller decrease was observed with the introduction of v6.2. As we will show below these are a direct consequence of algorithmic changes to the retrieved temperature of the surface and lower troposphere. Following these findings, the need arose for a self-consistent IASI NH$_3$ dataset, which uses stable and uniform

input data. The ECMWF ERA-Interim reanalysis (Dee et al., 2011) is very suitable for this purpose, as it provides all the necessary meteorological parameters and covers the whole IASI time period. We now detail how the ERA data are prepared for the neural network, and some of the assumptions that are made.

Two separate datasets are used: the first set consists of $0.25° \times 0.25°$ grids at a 3-hour temporal resolution and includes the following parameters: total water vapour column, surface pressure, temperature at 2 m and dew point temperature at 2 m.

The dew point temperature is used to calculate the specific humidity at the surface (Bolton, 1980). Temperature and specific humidity profiles are obtained at 37 pressure levels, on $0.25° \times 0.25°$ grids and at a 6-hour temporal resolution. For each IASI observation the closest grid cell is found, and for each parameter its value was linearly interpolated to the IASI overpass time. These parameters are then converted and/or interpolated to the required input format of the neural network.

Two parameters, namely cloud coverage and surface temperature, were judged to be too variable in space and time to rely

on gridded data. With respect to cloud coverage, we decided to continue to use the cloud information provided in the IASI L2 Eumetsat data, but instead of the standard 25 % threshold value, we now filter out all observations with a cloud coverage above 10 % (this leaves about 30 % of all observations for the latest version of IASI L2). Fortunately, except between 14 Sep 2010 and 28 Feb 2012 where a lot more observations were flagged as clear (see Table 2), the IASI L2 algorithm seems to be consistent enough for this approach to make sense. This choice also only affects data filtering, and not the retrieval itself. For

the surface temperature, this is not the case, and so the decision was made to setup a secondary neural network, dedicated to the retrieval of the surface temperature. The inputs of this neural network consist of 105 IASI channels, the satellite zenith angle and the emissivity. For the training and validation dataset, 37 days of the latest IASI L2 version (v6.2) ranging from June 2016 and June 2017 were used. Overall, the performance of this secondary NN yielded a standard deviation of 1.3 K and a mean surface temperature difference of -0.02 K (considering observations between $-60°$ and $60°$ of latitude). This in-house

retrieved surface temperature was therefore considered good enough to be used as a retrieval parameter in the neural network.

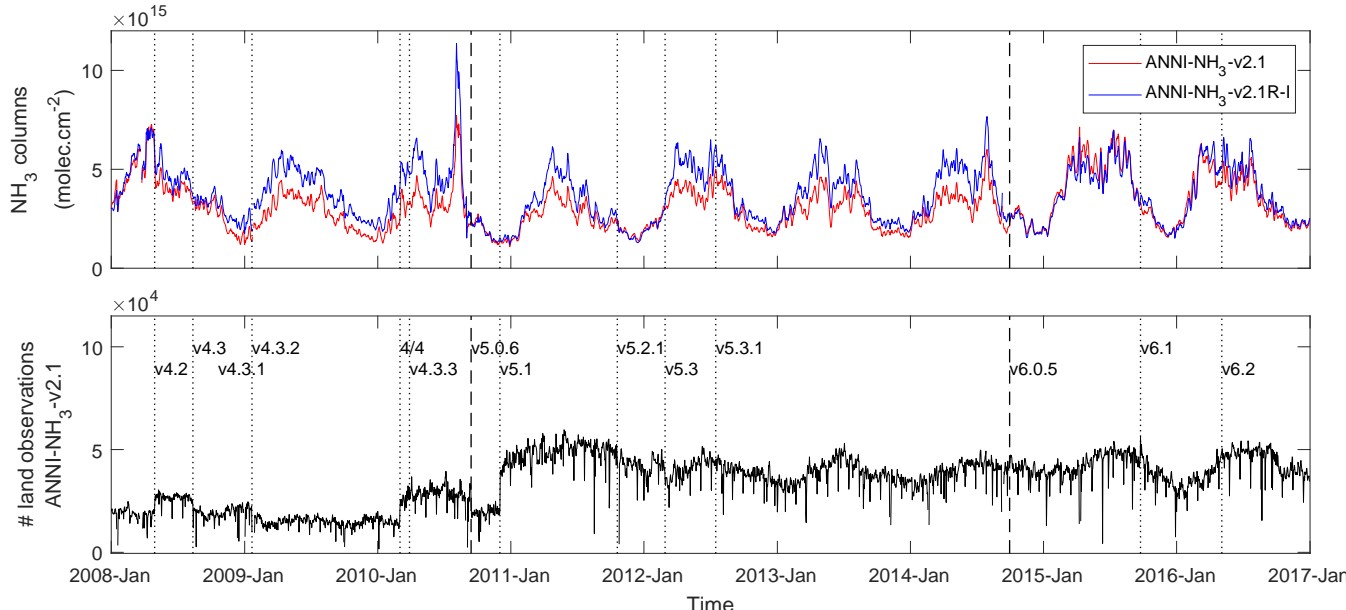

**Figure 3.** (Top) 5-day moving average timeseries of the morning $NH_3$ columns (molec.cm$^{-2}$) over the Northern Hemisphere for the near-real time retrieval (ANNI-$NH_3$-v2.1, red) and the reanalysed retrieval (ANNI-$NH_3$-v2.1R-I, blue). (Bottom) Number of land observations available for the Northern Hemisphere using the Eumetsat L2 data. The corresponding version number is indicated as a function of time.

Top panel of Figure 3 presents daily timeseries (5-day moving average) of the $NH_3$ columns for the reanalysed retrieval (ANNI-$NH_3$-v2.1R-I, blue), and the near-real time retrieval (ANNI-$NH_3$-v2.1, red) over the Northern Hemisphere. Figure 4 shows morning distributions over South Asia for 3 days corresponding to v5.3.1, v6.0.5 and v6.2 of the IASI Eumetsat L2 (see Table 2 and bottom panel of Figure 3). Taking the ANNI-$NH_3$-v2.1R-I as reference, it can be seen that prior to v6, retrieved

columns are much lower. With v6.0.5, the retrieved columns are slightly higher in magnitude. Finally, with v6.2 the retrieved columns are again a bit lower than the reanalysis, but still higher than with v5.3.1. From this, it can be deduced that the use of v6.0.5 resulted in a rather large increase of the $NH_3$ columns, while v6.2 resulted in a slight drop of the columns. Several different regions were studied, and these statements appear equally applicable elsewhere.

The observed biases can be attributed to changes in the IASI L2 retrieved temperature profile and surface temperatures,

as we will now demonstrate. Figure 5 shows standard deviations and mean temperature differences between IASI L2 v5.3.1, v6.0.5 and v6.2 and the reanalysis over land and sea (IASI observations between -60° and 60° of latitude on 29 September 2014, 1 October 2014 and 1 October 2016 respectively). These figures show both the temperature profile difference (coloured lines with respect to the ERA reanalysis) and the surface temperature difference (coloured crosses with respect to the NN retrieved surface temperature). From the standard deviation plots, it can be seen that the agreement of the L2 products with the

reanalysis improves with each version, except for the surface temperature of v6.0.5, which seemed to regress after v5.3.1 over

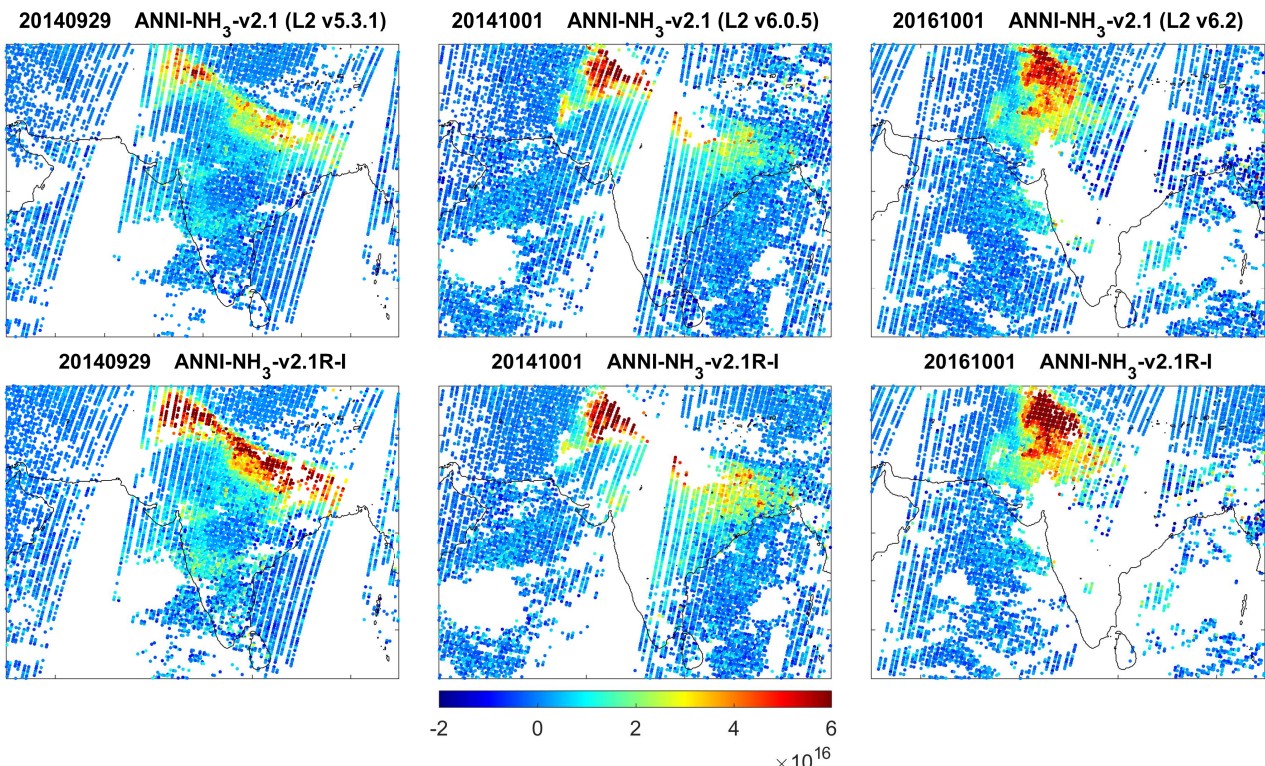

**Figure 4.** Example retrievals of the morning $NH_3$ column (molec.cm$^{-2}$) over South Asia for the reanalysed retrieval (ANNI-NH$_3$-v2.1R-I, bottom) versus the standard near-real time retrieval (ANNI-NH$_3$-v2.1, top) on 29 September 2014 (input data from Eumetsat L2 v5.3.1), 1 October 2014 (L2 v6.0) and 1 October 2016 (L2 v6.2).

land. Discontinuities and offset in the $NH_3$ product cannot be explained by changes in standard deviation, we therefore focus our attention to the mean difference.

We first discuss the observations over land. For the morning overpass of v5.3.1 we see a large negative offset of the air temperature in the lower troposphere (-5 K at the surface) and a much smaller negative offset in the surface temperature (-0.7

5    K). Overall, this implies that the IASI L2 v5.3.1 had an average high bias in the thermal contrast compared to the reanalysis, and therefore a low bias in retrieved $NH_3$ columns as was illustrated before with Figure 4. The air temperature offsets are highly reduced in v6.0.5 to about -2 K while the surface temperature offset regresses slightly to about the same -2 K value. The net result is that the thermal contrast decreases, and as consequence that an increase on average of the retrieved $NH_3$ columns is obtained. The fact that the offsets in the air and surface temperatures are almost identical, explains why the retrieved $NH_3$

10    columns of v2.1 with L2 v6.0.5 and the reanalysis are so similar. The main change in v6.2 was the improved surface temperature retrieval, which resolved the regression introduced in v6.0.5. However the overall increase in surface temperature causes the thermal contrast to increase, leading to lower retrieved $NH_3$ columns on average. The remaining offset in the surface air

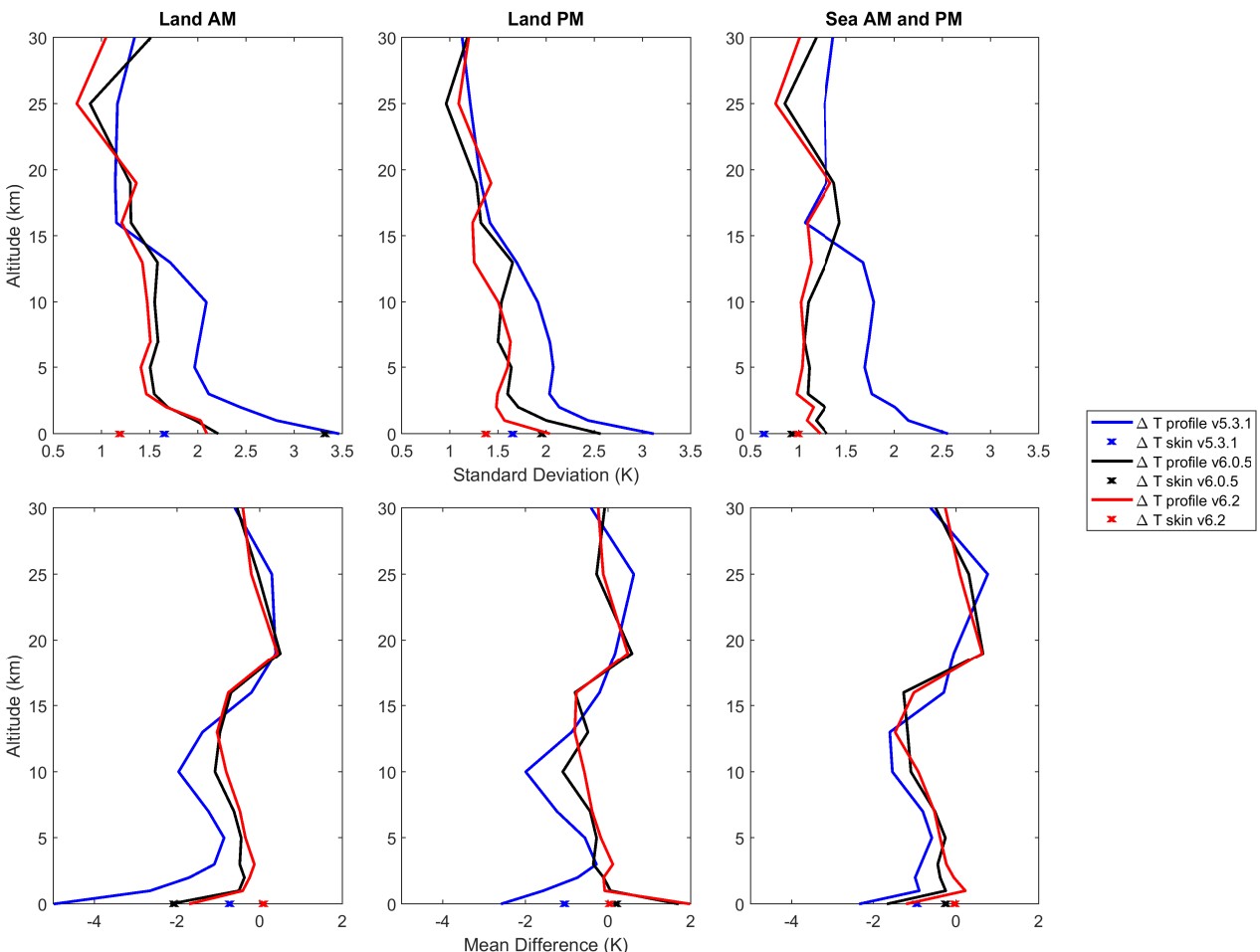

**Figure 5.** Standard deviations (top, K) and mean differences (bottom, K) of air and surface temperatures between IASI L2 v5.3.1 (29 September 2014), v6.0.5 (1 October 2014) and v6.2 (1 October 2016) and the reanalysis for the morning overpass over land (left), the evening overpass over land (middle) and the morning and evening overpasses over sea (right).

temperature (-1.7 K) implies that the ANNI-NH$_3$-v2.1 with the latest version of the L2 presents a low biased with respect to the reanalysis for morning observations over land. For the evening overpass, the air temperature of the IASI L2 v5.3.1 is characterised by a negative offset at the surface of -2.6 K, which combined with the negative mean difference of the surface temperature of -1.1 K results in a moderate high bias of the TC (1.5 K). In contrast, the air temperature offset becomes positive with v6 (and below 2 K) while the bias in surface temperature is close to 0, resulting therefore in a moderate low bias in TC. This implies, for land evening observations using L2 v5.3.1, a low bias of the v2.1 columns compared to the reanalysis; after version 6 of the L2, we find a high bias. Over sea, differences between the L2 datasets are smaller, both with respect to the surface temperatures and air temperature profiles, leading to smaller differences between the different products.

## 4  Concluding remarks

This paper presents the ANNI-NH$_3$-v2.1 retrieval, an improved version from the v1 detailed in Whitburn et al. (2016). The main changes are (1) a simplification of the input parameters and (2) the development of separate neural networks for land and sea observations, resulting in a better retrieval performance. As discontinuities are observed in the near-real time processing, a reanalysis of this version 2 was also introduced, namely the ANNI-NH$_3$-v2.1R-I which uses input generated from the ECMWF ERA-Interim dataset and a surface temperature retrieved by a secondary neural network. While further enhancements to the ANNI-NH$_3$ product are foreseen in the future (e.g., improved NH$_3$ columns could be achieved by using a distinct HRI for land and sea scene as input parameter), the neural network design described here is not expected to undergo major changes.

The presented analysis illustrates well the large impact that the (meteorological) input data can have on the retrieved NH$_3$ column. In particular, small absolute errors in the TC can lead to very large inaccuracies on the retrieved columns, especially when the TC itself is small. Incremental improvements in IASI L2 temperature and cloud algorithms and/or ECMWF ERA-Interim data are therefore expected to have a positive impact on the quality of the NH$_3$ datasets. While certain biases might still be present, the ANNI-NH$_3$-v2.1R-I is self-consistent in time and is expected to be highly suitable to study long-term trends.

*Data availability.*  The near-real time ANNI-NH$_3$-v2.1 data used in this work are freely available for all users through the AERIS database http://iasi.aeris-data.fr/NH3/. The ANNI-NH$_3$-v2.1R-I dataset will also be made available at the same place and its delivery is planned for the beginning of 2018.

*Competing interests.*  No competing interests are present.

*Acknowledgements.*  IASI has been developed and built under the responsibility of the Centre National d'Études spatiales (CNES, France). It is flown on board the Metop satellites as part of the EUMETSAT Polar System. The IASI L1c data are received through the EUMETCast near real-time data distribution service. The research was funded by the F.R.S.-FNRS and the Belgian State Federal Office for Scientific, Technical and Cultural Affairs (Prodex arrangement IASI.FLOW) and EUMETSAT/AC-SAF project. S. Whitburn is grateful for his Ph.D. grant (Boursier FRIA) to the "Fonds pour la Formation à la Recherche dans l'Industrie et dans l'Agriculture" of Belgium. L. Clarisse is Research Associate (Chercheur Qualifié) with the Belgian F.R.S.-FNRS. C. Clerbaux is grateful to CNES for scientific collaboration and financial support.

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
