# Peer review of "Version 2 of the IASI NH3 neural network retrieval algorithm; near-real time and reanalysed datasets"

_Atmospheric Measurement Techniques, 2017_

## Referee Comment (RC1) · Anonymous Referee #2 · 21 Aug 2017

Van Damme et al. present the version 2 of the IASI NH3 neural network retrieval algorithm (ANNI-NH3-v2). This version is an improved version of previous developed and published, version 1 (Whitburn et al., JGR, 2016). The main improvements concern: separated land and sea neural networks, extended training dataset (including more representative scenes for high thermal contrasts and low latitudes), and change in the NN output to avoid overtraining the low sensitivity scenes. Thanks to the expertise acquired with the previous NH3 algorithms developed by the ULB group, improvements and simplifications of the input parameters have been made as well as for the post-filtering process. Performances of the new version are compared to the previous version and recommendations for use are made. Finally, the authors present another

version of the ANNI-NH3-v2, ANNI-NH3-V2R-I. This version allows the correction of errors and biases introduced by changes in the EUMETSAT version of meteorological parameters needed for the NH3 retrieval. These changes introduce discontinuities in the NH3 timeseries. To avoid this, the authors based their retrievals on the ECMWF ERA-Interim reanalysis for temperature profiles and the development of a NN for surface temperature retrieval. They discuss the implication of the changes in the meteorological parameters on the NH3 retrieval and recommend the use of ANNI-NH3-v2R-I for long timeseries analyses, when the product will be available. The paper is well written and structured with detailed discussions of the major changes in the algorithm and their implications in terms of NH3 retrieval for both the versions ANNI-NH3-v2 and ANNI-NH3-v2R-I. This work is suitable for AMT publication and the recommendations made for the use of the different products is very useful for the users, especially considering the warning and improvements for the long timeseries. I recommend this paper for publication in AMT after the following comments are addressed. - The authors should consider introducing a table summarizing the changes between the 2 versions 1 and 2 and listing the different inputs parameters and their description used in the NN (page 5). - Page 7, lines27-33: it would be interesting to show series for one or more regions to illustrate the discontinuities and to show how the ANNI-NH3-v2R-I reduces these discontinuities. - In the data availability statement, it would be useful for the readers/users to have the information about the period from when the authors expect that the new data will be publicly available. Technical comments: - Page 2, line 20-21: "detailed" and "introduced" should be inverted.

---

## Referee Comment (RC2) · Anonymous Referee #1 · 31 Aug 2017

REFEREE COMMENTS on

'Version 2 of the IASI NH3 neural network retrieval algorithm; near-real time and re-analysed datasets' by van Damme et al.

SUMMMARY

The paper describes updates to an operational near-real time neural network algorithm to retrieve NH3 from IASI measurements, as well as an alternative product based on ECMWF reanalyses which produces a more stable long-term dataset. This follows on from an earlier publication in JGR (Whitburn et al, 2016) which gave a detailed description of the original algorithm, including some validation.

[Figure]

The paper itself seems more of an appendix to the the original paper, which is unfortunately in a different journal (JGR, although apparently open access) and might therefore have made more sense if this paper was also submitted to JGR rather than AMT. However, since this paper describes the algorithms used to generate a publicly available dataset, it should be published somewhere.

GENERAL COMMENTS

1) I understand the authors not wishing simply to repeat much of what was written in the original paper, however I feel just a few extra lines describing atmospheric NH3, the IASI instrument and the concepts of HRI and the neural network approach, as part of the introduction, would help make the paper stand alone as an independent publication (especially given the change of journal). On the other hand, the algorithm changes themselves are actually quite well explained and I didn't feel the need to read the original paper to understand those parts.

2) Despite reading the original paper, I'm still somewhat confused as to what the 'error' associated with the NH3 product is supposed to represent (which also arises from the point made on p7 L14). As I understand it there are two possible contributions. Firstly, there is the propagation of the instrument noise and various systematic error terms through the covariance matrix in the HRI component. Secondly there is the residual error from the neural network fit (for example, as shown in Fig 1), which may, or may not, be represented as an error in the scaling factor f.

3) As well as the maps in Fig 2 and Fig 3 it would be useful to have extra panels showing, for the common locations, a scatter plot of the differences between the retrievals in question. Also, relating to point 2, whether such differences can be explained in terms of the associated errors assigned to the product (bearing in mind that many such error components will cancel out when changing just the algorithm rather than the measurements). Section 3 contains a fairly lengthy qualitative discussion on how changes in the input data (surface and air temperature) lead to expected changes in sign in the

retrieved NH3, but it seems this could quite easily be quantified via linear error propagation of the mean difference profiles shown in Fig 4 through the HRI and application of the neural network for the associated scaling factor.

SPECIFIC COMMENTS

a) P5 L4: It would be useful to have a list of these 20 input parameters.

b) P5 L11: Of course, if the simulated spectra fail to take account of the proper treatment of surface reflections at high angles of incidence then there will always be an issue with real data at such angles. Could this be part of the problem? Certainly high angles seem to problematic for all retrievals, not just a neural-network problem.

c) P5 L14: by 'cosine of the viewing angle' I suspect you mean 'secant of the zenith angle' - the viewing angle, as measured from the satellite, generally relating less well to air mass than the zenith angle as measured on the ground once the earth's curvature is taken into consideration.

d) P7 L14: Even unweighted averages can be skewed by outliers. Have you considered simply recommending using a median value instead?

f) P7 L27: Is there some simple way to demonstrate these discontinuities in the time series (eg a plot of the median value over a large area) and, equally, the resulting 'improvement' from using reanalysis data?

TECHNICAL CORRECTIONS

P4 Eq 1: f not defined.

P6 L7: 'built' rather than 'build'.

P6 L22: 'third criterion' rather than 'third criteria'.

P7 L16: 'retrieval algorithms' rather than 'retrieval algorithm'.

P7 L19: (pedantically) 'data were' rather than 'data was'.

---

## Author Comment (AC1) · 20 Oct 2017

**Anonymous Referee #1 Received and published: 31 August 2017**

We would like to thank the referee for the constructive remarks and comments on this manuscript. Taking into account them greatly improved the manuscript. The point-by-point responses are provided below in blue.

We also made two additional changes:

1. During the discussion period, further analyses have demonstrated that the bias correction applied over sea should also be applied over land. This was shown by the negative biases observed over islands in the tropics. Applying this correction improves the obtained distributions. We therefore describe this in the manuscript by changing the paragraph (P6, L14-16 and P7, L1-2, numbering of updated manuscript) to "A final change introduced in v2 is a HRI bias correction over the seas, where the HRI was found to be slightly negative overall, and decreasing with increasing H2O total column amount. As this was identified to be also the case over land, the same correction was applied over land in v2.1. A H2O dependent bias was determined from a region assumed NH3-free by calculating the median over sea for 30 days in 2015 over bins of  $0.1 \cdot 10^{23}$  molec.cm-2 of H2O total column. These median values are then used to correct the HRIs before using them as an input in the neural network."

All the figures have been adapted following this change and now present the ANNI-NH3-v2.1 dataset. Additional textual changes to v2.1 have been done all over the manuscript.

2. A correction was made P9 L32-33 where the sentence "The inputs of this neural network consist of 105 IASI channels and the satellite zenith angle." has been changed to "The inputs of this neural network consist of 105 IASI channels, the satellite zenith angle and the emissivity."

REFEREE COMMENTS on 'Version 2 of the IASI NH3 neural network retrieval algorithm; near-real time and reanalysed datasets' by van Damme et al.

**SUMMMARY**

The paper describes updates to an operational near-real time neural network algorithm to retrieve NH3 from IASI measurements, as well as an alternative product based on ECMWF reanalyses which produces a more stable long-term dataset. This follows on from an earlier publication in JGR (Whitburn et al, 2016) which gave a detailed description of the original algorithm, including some validation.

The paper itself seems more of an appendix to the the original paper, which is unfortunately in a different journal (JGR, although apparently open access) and might therefore have made more sense if this paper was also submitted to JGR rather than AMT. However, since this paper describes the algorithms used to generate a publicly available dataset, it should be published somewhere.

**GENERAL COMMENTS**

1) I understand the authors not wishing simply to repeat much of what was written in the original paper, however I feel just a few extra lines describing atmospheric NH3, the IASI instrument and the concepts of HRI and the neural network approach, as part of the introduction, would help make the paper stand alone as an independent publication (especially given the change of journal). On the other hand, the algorithm changes themselves are actually quite well explained and I didn't feel the need to read the original paper to understand those parts.

As noted by the referee, we made some effort to make this paper readable by itself. It was a redaction choice to avoid description of the IASI instrument and the retrieval method used which are thoroughly described in other papers (especially in Van Damme et al. (2014) and Whitburn et al. (2016)). The focus was made on the algorithm improvements and the impact of the input data on the retrieved columns.

2) Despite reading the original paper, I'm still somewhat confused as to what the 'error' associated with the NH3 product is supposed to represent (which also arises from the point made on p7 L14). As I understand it there are two possible contributions. Firstly, there is the propagation of the instrument noise and various systematic error terms through the covariance matrix in the HRI component. Secondly there is the residual error from the neural network fit (for example, as shown in Fig 1), which may, or may not, be represented as an error in the scaling factor f.

The "error" represents how the uncertainty of each input parameter of the NN propagates to the retrieved column. In other words, it evaluates how a variation of the parameters will affect the retrieved column.

The uncertainty associated with the HRI (equal to one by definition since the HRI of spectra without NH3 have a mean of 0 and a standard deviation of 1) is mainly caused by the random instrumental noise. For the other parameters, the error terms are set using uncertainty taken from earlier validation of the IASI Level 2 meteorological fields.

The residual error or bias from the neural network fit is here not taken into account in the calculation of the error on retrieved column from real IASI observations.

3) As well as the maps in Fig 2 and Fig 3 it would be useful to have extra panels showing, for the common locations, a scatter plot of the differences between the retrievals in question. Also, relating to point 2, whether such differences can be explained in terms of the associated errors assigned to the product (bearing in mind that many such error components will cancel out when changing just the algorithm rather than the measurements). Section 3 contains a fairly lengthy qualitative discussion on how changes in the input data (surface and air temperature) lead to expected changes in sign in the retrieved NH3, but it seems this could quite easily be quantified via linear error propagation of the mean difference profiles shown in Fig 4 through the HRI and application of the neural network for the associated scaling factor.

As mentioned in point (2), the "error" associated to the retrievals is calculated by propagating the uncertainty of the different parameters used as input by the neural network. As the parameters driving the error (mainly HRI and temperature profile) are identical between the NN-v1 and NN-v2.1, the error is similar. What drives the changes is the better performance of the NN, which is determined by the improved set of simulations and the better coverage of the range of thermal contrasts.

Figure 1: Scatter plots of the ANNI-NH3-v1 versus ANNI-NH3-v2 (molec/cm2). The color scale represents the error of v1 (left) and v2 (right).

The error does not vary significantly from one dataset to another in contrast with the retrieved column (see Figure 1). If this was the case, the more important changes (in relative value) on the retrievals would be observed for low columns while high columns would be less impacted (since the  $NH_3$  signal in the IASI spectrum is better and the influence of small changes in the value of the input parameters are therefore less pronounced). What we observe is the opposite, with the biggest changes found for the highest retrieved columns. We made the choice to not add these scatter plots in the manuscript, as they do not provide substantial information. However, we have added a sentence P7 L21-23 to clarify this: "It is worth noting that the uncertainty associated with v1 and with v2.1 does not vary substantially for source regions, as it is mainly driven by the HRI and the temperature profile".

Figure 4 (5 in the updated manuscript) presents case studies for specific days. We cannot derive a mean difference profile representative for difference between meteorological L2 data and ECMWF data. Similarly, we cannot deduce a mean profile used by the NN.

**SPECIFIC COMMENTS**

**a) P5 L4: It would be useful to have a list of these 20 input parameters.**

**We agree with the referee and have added a table listing all the changes between v1 and v2.1 (Table 1, P6 and here below) which includes the 20 input parameters of v2.1.**

|                    | NN-v1                                                                                     | NN-v2.1                                                                                       |
|--------------------|-------------------------------------------------------------------------------------------|-----------------------------------------------------------------------------------------------|
|                    | [NH 3 ]                                                                        | HRI                                                                                           |
| Output parameter   | HRI                                                                                       | $\overline{[\mathrm{NH}_3]}$                                                                  |
| Input parameters*  | 31: T (12 levels), $T_{surf}$ , P (11 levels), $H_2O$ (7 levels),                         | 20: T (12 levels), $T_{surf}$ , $P_{surf}$ , $H_2O$ total column, $\sigma$ ,                  |
|                    | $\sigma$ , $z_0$ , $\epsilon$ , angle                                                     | $z_0, \epsilon$ , angle, HRI                                                                  |
| Training set       | 250000 simulations                                                                        | 450000 simulations                                                                            |
| Land/sea treatment | One network                                                                               | Separate networks                                                                             |
| Angle treatment    | Angle dependent HRIs                                                                      | $1^{st}$ order correction of the HRIs by the cosine of the                                    |
|                    |                                                                                           | zenith angle                                                                                  |
|                    |                                                                                           | Angle as input parameter for $2^{nd}$ order corrections                                       |
| Bias correction    | No                                                                                        | Over sea (v2 dataset) / Over land and sea (v2.1 dataset)                                      |
| Pre-filtering**    | Cloud cover $> 25 \%$                                                                     | Cloud cover $> 25 \%$                                                                         |
| Post-filtering**   | $[NH_3] < 0$ and HRI > 1.5 in absolute value                                              | $[NH_3] < 0$ and HRI $> 1.5$ in absolute value                                                |
|                    | $\frac{[\rm NH_3]}{\rm HRI} > 3\cdot 10^{16} \ \rm molec.cm^{-2} \ in \ absolute \ value$ | $\frac{[\rm NH_3]}{\rm HRI} > 1.75 \cdot 10^{16} \ \rm molec.cm^{-2} \ in \ absolute \ value$ |

Table 1. List of changes from ANNI-NH3-v1 to ANNI-NH3-v2.1.

 $*\sigma$  and  $z_0$  are parameters characterizing the shape of the NH3 vertical profile;  $\epsilon$  represents the emissivity.

\*\*An observation is removed as soon as one of the criteria is met.

b) P5 L11: Of course, if the simulated spectra fail to take account of the proper treatment of surface reflections at high angles of incidence then there will always be an issue with real data at such angles. Could this be part of the problem? Certainly high angles seem to problematic for all retrievals, not just a neural-network problem.

**In our case, what we want to correct is the higher signal to noise ratio at high viewing angle. It is a bias inherent to this HRI method and not related to radiative transfer issue at high angle.**

c) P5 L14: by 'cosine of the viewing angle' I suspect you mean 'secant of the zenith angle'-the viewing angle, as measured from the satellite, generally relating less well to air mass than the zenith angle as measured on the ground once the earth's curvature is taken into consideration.

Thank you for pointing this out. We indeed used the cosine of the zenith angle (the text has been updated accordingly P6 L2 & L11). This is a first order correction on the HRI and was found to work better than with the satellite viewing angle. However, any remaining angle dependency is addressed by adding the zenith angle to the network. It has been clarified in the text P6 L11-12: "Note that the zenith angle is still kept as a parameter in the neural network, to allow the neural network to perform second order corrections to address any remaining angle dependency."

d) P7L14: Even unweighted averages can be skewed by outliers. Have you considered simply recommending using a median value instead?

Thank you for this suggestion. We have added in the manuscript. P8 L4-5: "Using median values is also an option to decrease the importance of outliers."

f) P7 L27: Is there some simple way to demonstrate these discontinuities in the time series (eg a plot of the median value over a large area) and, equally, the resulting 'improvement' from using reanalysis data?

We agree with the referee and to show the discontinuities, we have added an additional figure (Figure 3, P10 and here below). It presents ANNI-NH3-v2.1 and ANNI-NH3-v2.1R-I timeseries over the Northern Hemisphere and clearly shows the added value of the reanalysis. While an increase of NH3 columns is observed in 2015 with the introduction of the meteorological L2 in ANNI-NH3-v2.1 dataset, the ANNI-NH3-v2.1R-I dataset presents a more consistent NH3 column record over time.

**Figure 3.** (Top) 5-day moving average timeseries of the morning  $NH_3$  columns (molec.cm-2) over the Northern Hemisphere for the nearreal time retrieval (ANNI-NH3-v2.1, red) and the reanalysed retrieval (ANNI-NH3-v2.1R-I, blue). (Bottom) Number of land observations available for the Northern Hemisphere using the Eumetsat L2 data. The corresponding version number is indicated as a function of time.

**We updated the text P9 (updated manuscript) and the first paragraph now reads:**

"Top panel of Figure 3 presents daily timeseries (5-day moving average) of the NH3 columns for the reanalysed retrieval (ANNI-NH3-v2.1R-I, blue), and the near-real time retrieval (ANNI-NH3-v2.1, red) over the Northern Hemisphere. Figure 4 shows morning distributions over South Asia for 3 days corresponding to v5.3.1, v6.0.5 and v6.2 of the IASI Eumetsat L2 (see Table 2 and bottom panel of Figure 3). Taking the ANNI-NH3-v2.1R-I as reference, it can be seen that prior to v6, retrieved columns are much lower. With v6.0.5, the retrieved columns are slightly higher in magnitude. Finally, with v6.2, the retrieved columns are again a bit lower than the reanalysis, but still higher than with v5.3.1. From this, it can be deduced that the use of v6.0.5 resulted in a rather large increase of the NH3 columns, while v6.2 resulted in a slight drop of the columns. Several different regions were studied, and these statements appear equally applicable elsewhere."

A new sentence was also added P9 L2-3: "This is illustrated for land observations over the Northern Hemisphere in the bottom panel of Figure 3."

**TECHNICAL CORRECTIONS**

We thank the anonymous referee to have taken his time to list these technical corrections.

P4 Eq 1: f not defined.

f represents the ratio, this has been clarified in the text P4 L6.

P6 L7: 'built' rather than 'build'.

It has been corrected accordingly.

P6 L22: 'third criterion' rather than 'third criteria'.

It has been corrected accordingly.

P7 L16: 'retrieval algorithms' rather than 'retrieval algorithm'.

It has been corrected accordingly.

P7 L19: (pedantically) 'data were' rather than 'data was'.

It has been corrected accordingly (2 times in this paragraph). To be consistent we also changed:

"The relevant L2 data consists..." to "The relevant L2 data consist..."

"We now detail how the ERA data is..." to" We now detail how the ERA data are..."

Table 1: 30 Sep 2014. "IASI meteorological L2 data is..." to "IASI meteorological L2 data are..."

---

## Author Comment (AC2) · 20 Oct 2017

We would like to thank the anonymous referee for the review and comments on the manuscript, which was greatly improved after considering them. The point-by-point responses are provided below in blue.

We also made two additional changes:

1. During the discussion period, further analyses have demonstrated that the bias correction applied over sea should also be applied over land. This was shown by the negative biases observed over islands in the tropics. Applying this correction improves the obtained distributions. We therefore describe this in the manuscript by changing the paragraph (P6, L14-16 and P7, L1-2, numbering of updated manuscript) to "A final change introduced in v2 is a HRI bias correction over the seas, where the HRI was found to be slightly negative overall, and decreasing with increasing $H_2O$ total column amount. As this was identified to be also the case over land, the same correction was applied over land in v2.1. A $H_2O$ dependent bias was determined from a region assumed $NH_3$-free by calculating the median over sea for 30 days in 2015 over bins of $0.1 \cdot 10^{23}$ molec.cm$^{-2}$ of H2O total column. These median values are then used to correct the HRIs before using them as an input in the neural network."
   All the figures have been adapted following this change and now present the ANNI-NH$_3$-v2.1 dataset. Additional textual changes to v2.1 have been done all over the manuscript.

2. A correction was made P9 L32-33 where the sentence "The inputs of this neural network consist of 105 IASI channels and the satellite zenith angle." has been changed to "The inputs of this neural network consist of 105 IASI channels, the satellite zenith angle and the emissivity."

Van Damme et al. present the version 2 of the IASI NH3 neural network retrieval algorithm (ANNI-NH3-v2). This version is an improved version of previous developed and published, version 1 (Whitburn et al., JGR, 2016). The main improvements concern: separated land and sea neural networks, extended training dataset (including more representative scenes for high thermal contrasts and low latitudes), and change in the NN output to avoid overtraining the low sensitivity scenes. Thanks to the expertise acquired with the previous NH3 algorithms developed by the ULB group, improvements and simplifications of the input parameters have been made as well as for the post-filtering process. Performances of the new version are compared to the previous version and recommendations for use are made. Finally, the authors present another version of the ANNI-NH3-v2, ANNI-NH3-V2R-I. This version allows the correction of errors and biases introduced by changes in the EUMETSAT version of meteorological parameters needed for the NH3 retrieval. These changes introduce discontinuities in the NH3 timeseries. To avoid this, the authors based their retrievals on the ECMWF ERA-Interim reanalysis for temperature profiles and the development of a NN for surface temperature retrieval. They discuss the implication of the changes in the meteorological parameters on the NH3 retrieval and recommend the use of ANNI-NH3-v2R-I for long timeseries analyses, when the product will be available. The paper is well written and structured with detailed discussions of the major changes in the algorithm and their implications in terms of NH3 retrieval for both the versions ANNI-NH3-v2 and ANNI-NH3-v2R-I. This work is suitable for AMT publication and the recommendations made for the use of the different products is very useful for the users, especially considering the warning and improvements for the long timeseries. I recommend this paper for publication in AMT after the following comments are addressed.

- The authors should consider introducing a table summarizing the changes between the 2 versions 1 and 2 and listing the different inputs parameters and their description used in the NN (page 5).

We agree with the referee and have added a table listing all the changes between v1 and v2.1 (Table 1, P6 and here below) which includes the 20 input parameters of v2.1.

**Table 1.** List of changes from ANNI-NH$_3$-v1 to ANNI-NH$_3$-v2.1.

| | NN-v1 | NN-v2.1 |
|---|---|---|
| Output parameter | $\dfrac{[NH_3]}{HRI}$ | $\dfrac{HRI}{[NH_3]}$ |
| Input parameters* | 31: T (12 levels), $T_{surf}$, P (11 levels), $H_2O$ (7 levels), $\sigma$, $z_0$, $\epsilon$, angle | 20: T (12 levels), $T_{surf}$, $P_{surf}$, $H_2O$ total column, $\sigma$, $z_0$, $\epsilon$, angle, HRI |
| Training set | 250000 simulations | 450000 simulations |
| Land/sea treatment | One network | Separate networks |
| Angle treatment | Angle dependent HRIs | $1^{st}$ order correction of the HRIs by the cosine of the zenith angle. Angle as input parameter for $2^{nd}$ order corrections |
| Bias correction | No | Over sea (v2 dataset) / Over land and sea (v2.1 dataset) |
| Pre-filtering** | Cloud cover > 25 % | Cloud cover > 25 % |
| Post-filtering** | $[NH_3] < 0$ and HRI > 1.5 in absolute value $\dfrac{[NH_3]}{HRI} > 3 \cdot 10^{16}$ molec.cm$^{-2}$ in absolute value | $[NH_3] < 0$ and HRI > 1.5 in absolute value $\dfrac{[NH_3]}{HRI} > 1.75 \cdot 10^{16}$ molec.cm$^{-2}$ in absolute value |

*$\sigma$ and $z_0$ are parameters characterizing the shape of the NH$_3$ vertical profile; $\epsilon$ represents the emissivity.

**An observation is removed as soon as one of the criteria is met.

- Page 7, lines27-33: it would be interesting to show series for one or more regions to illustrate the discontinuities and to show how the ANNI-NH3-v2R-I reduces these discontinuities.

We agree with the referee and have added a figure (Figure 3, P10 and here below) to illustrate the discontinuities. It presents ANNI-NH3-v2.1 (red) and ANNI-NH3-v2.1R-I (blue) timeseries over the Northern Hemisphere and clearly shows the added value of the reanalysis. While an increase of NH$_3$ columns is observed in 2015 with the introduction of the meteorological L2 in ANNI-NH$_3$-v2.1 dataset, the ANNI-NH3-v2.1R-I dataset present a more consistent NH3 column record over time.

[Figure]

**Figure 3.** (Top) 5-day moving average timeseries of the morning $NH_3$ columns (molec.cm$^{-2}$) over the Northern Hemisphere for the near-real time retrieval (ANNI-NH$_3$-v2.1, red) and the reanalysed retrieval (ANNI-NH$_3$-v2.1R-I, blue). (Bottom) Number of land observations available for the Northern Hemisphere using the Eumetsat L2 data. The corresponding version number is indicated as a function of time.

The first paragraph of P9 (updated manuscript) now reads:

"Top panel of Figure 3 presents daily timeseries (5-day moving average) of the NH$_3$ columns for the reanalysed retrieval (ANNI-NH$_3$-v2.1R-I, blue), and the near-real time retrieval (ANNI-NH$_3$-v2.1, red) over the Northern Hemisphere. Figure 4 shows morning distributions over South Asia for 3 days corresponding to v5.3.1, v6.0.5 and v6.2 of the IASI Eumetsat L2 (see Table 2 and bottom panel of Figure 3). Taking the ANNI-NH$_3$-v2.1R-I as reference, it can be seen that prior to v6, retrieved columns are much lower. With v6.0.5, the retrieved columns are slightly higher in magnitude. Finally, with v6.2, the retrieved columns are again a bit lower than the reanalysis, but still higher than with v5.3.1. From this, it can be deduced that the use of v6.0.5 resulted in a rather large increase of the NH$_3$ columns, while v6.2 resulted in a slight drop of the columns. Several different regions were studied, and these statements appear equally applicable elsewhere."

A new sentence was also added P9 L2-3: "This is illustrated for land observations over the Northern Hemisphere in the bottom panel of Figure 3."

 - In the data availability statement, it would be useful for the readers/users to have the information about the period from when the authors expect that the new data will be publicly available.

The data availability statement has been changed to: "The near-real time ANNI-NH$_3$-v2.1 data used in this work are freely available for all users through the AERIS database http://iasi.aeris-data.fr/NH3/. The ANNI- NH$_3$-v2.1R- dataset will also be made available at the same place and its delivery is planned for the beginning of 2018."

Technical comments: - Page 2, line 20-21: "detailed" and "introduced" should be inverted.

This has been corrected accordingly.